# The Immunological Changes in the Skin of BALC/c Mice with Disseminated Acanthamoebiasis

**DOI:** 10.3390/pathogens12050631

**Published:** 2023-04-22

**Authors:** Agnieszka Wojtkowiak-Giera, Monika Derda, Natalia Łanocha-Arendarczyk, Agnieszka Kolasa, Karolina Kot, Joanna Walczykiewicz, Piotr Solarczyk, Danuta Kosik-Bogacka

**Affiliations:** 1Department of Biology and Medical Parasitology, Poznan University of Medical Sciences, 60-781 Poznan, Poland; 2Department of Biology and Medical Parasitology, Pomeranian Medical University in Szczecin, 70-111 Szczecin, Poland; 3Department of Histology and Embryology, Pomeranian Medical University in Szczecin, 70-111 Szczecin, Poland; 4Independent of Pharmaceutical Botany, Pomeranian Medical University in Szczecin, 70-111 Szczecin, Poland

**Keywords:** acanthamoebiasis, mice, quantitative real-time PCR, Toll-like receptor 2, Toll-like receptor 4

## Abstract

Toll-like receptors (TLR) are involved in the recognition of numerous pathogens, including *Acanthamoeba* spp. Thanks to this, it is possible for immune cells to recognize microorganisms and trigger the body’s innate immune response. The stimulation of TLRs also leads to the activation of specific immunity. The aim of the study was to determine the TLR2 and TLR4 gene expression in the skin of BALC/c mice infected with *Acanthamoeba* with AM22 strain isolated from a patient. Receptor expression was assessed by real-time polymerase chain reaction (qPCR) in the amoeba-infected host with normal (A) and reduced immunity (AS) as well as in the control host with normal immunity (C) and reduced immunity (CS). Statistical analysis of TLR2 gene expression in A and AS groups compared to C and CS groups, respectively, were statistically insignificant. In the A group, we found statistical upregulation of TLR4 gene expression at 8 dpi compared to the C group. While in AS group, TLR4 gene expression was at a similar level, such as in the CS group. Taking into account the host’s immune status, the TLR4 gene expression was statistically higher in the skin of host from A group than in host from AS group at the beginning of the infection. Increased TLR4 gene expression in hosts with normal immunity infected with *Acanthamoeba* suggests the involvement of the studied receptor in the course of acanthamoebiasis. The above research results provide new data on the involvement of the studied receptor in the skin in the host’s immune defense triggered during the *Acanthamoeba* infection.

## 1. Introduction

Free-living amoebae of the genus *Acanthamoeba* are cosmopolitan protozoans and pathogenic strains that may cause infections in the brain (granulomatous amoebic encephalitis, GAE), eye (*Acanthamoeba* keratitis, AK), lungs and skin. Cutaneous amoebiasis (CA) is a rare opportunistic infection; primary skin amoebic changes as well as cutaneous lesions from disseminated amoebiasis are much more common in patients with *immunodeficiency* disorders, including patients with an HIV infection and patients on immunosuppressive therapy with or without CNS involvement [1]. Risk factors for *Acanthamoeba* skin lesions include traumatized areas, such as surgical scars, viral lesions, bites and mechanical trauma [2,3]. *Acanthamoeba* spp. skin infections may be the manifestation of an infection that is initiated in another organ and spreads hematogenously or are the primary focus of infection through skin wounds that posteriorly spread to other tissues [4]. As the first observed symptom of skin *Acanthamoeba* spp. infection wound inflammation develops, it results in skin changes [5]. Cutaneous acanthamoebiasis can present variable lesions, including papules, pustules, nodules and skin ulcers that contain *Acanthamoeba* trophozoites and cysts. Some authors observed that *Acanthamoeba* inducted skin tissue necrosis [6,7,8,9].

Toll-like receptors (TLRs) are a family of protein-transmembrane receptors. TLR2 and TLR4 are the most extensively investigated members of the family. Our earlier studies confirmed changes in the expression of TLR2 and TLR4 in organs, such as kidneys, heart and eyes of mice infected with *Acanthamoeba*. Therefore, we selected the same receptors to examine in the skin of immunocompetent and immunosuppressed mice post-*Acanthamoeba* spp. infection [10,11,12,13]. Additionally, TLR4 is the receptor that stimulates the pathways of TLR4-MyD88-NF-κB and TLR4-ERK1/2 and activates inflammatory cytokines in acanthamoebiasis [7,14,15]. As well, TLR plays an important role in directing the T helper cell (Th) differentiation during amoeba infection [16].

These receptors induce the production of inflammatory mediators after recognition of pathogen-associated molecular patterns (PAMP) or host-derived damage-associated molecular patterns (DAMPs) [17,18,19].

One of the first described receptors in this family was TLR2 (transmembrane protein) which recognizes a wide variety of ligands. The best-known ligands in parasites, such as *Trypanosoma cruzi*, *T. brucei*, *Plasmodium falciparum*, *Toxoplasma gondii*, *Leishmania donovani* and *L. major* are glycosylphosphatidylinositol (GPI) anchors and lysophosphatidylserine found in *Schistosoma mansoni* [20,21,22,23,24]. In addition, TLR2 shows the ability to form complexes with the receptor TLR1 and TLR6, which increase the range of recognized ligands [25]. The second extracellular receptor, of which the best-known ligand is LPS (lipopolysaccharide), present in the cell wall of Gram-negative bacteria is TLR4. Other ligands include heat shock protein (HSP60, HSP70 and Cp96) and lipophosphoglycan (LPG) found in *Leishmania* spp., lysophosphatidylserine, present in *Schistosoma* spp., lipophosphoglycan in *L. major*, lipopeptidophosphoglycan in *Entamoeba histolytica*, phospholipid in *Ascaris lumbricoides* [26,27,28,29,30] and GPI occurring in *Plasmodium falciparum*, *Toxoplasma gondii* and *T. cruzi* [31].

The signal transduction process occurs as a result of contact with pathogens and recognition of different ligands by TLRs. The adapter proteins involved in this process are TIRAP (TIR-domain-containing protein), TRIF (TIR-domain-containing adapter inducing INF-β), TRAM (TRIF-related adapter molecule) and MyD88 (myeloid differentiation factor 88). The signal transduction can be divided into two pathways: protein Myd88 dependent and independent [32]. The protein MyD88 is involved in TLR2-mediated signaling in the cell which binds to the receptor via the protein adapter TIRAP. This is followed by the activation of IRAK-4 and autophosphorylation of IRAK-1 which is released from the complex with the receptor and binds to the TRAF-6 factor [33]. In the next step, the complex TAK-1-TAB activates kinases (IKK and IKKβ) and stimulates transcription factor NF-κB, stimulating the expression of proinflammatory cytokines, such as IL-6, IL-10 and TNF-α [34].

In the case of TLR4 signaling transduction, there is the participation of two protein adapters, MyD88 and TRIF. The pathway with the TRIF protein adapter requires involvement of the second protein adapter, TRAM. This connects two adapter-activated kinases TBK1 (TANK-Binding Kinase-1) and IKKε/IKKi. The next step is the activation of the transcription factor IRF3 which stimulates interferon I gene expression and the production of interleukins, such as IL-1, Il-6, Il-8, Il-12 and Il-18 [31,35,36]. The signal transduction is also affected by the protein Tollip (Toll interacting protein). This protein acts as an IRAk-1 kinase to block its action [37].

No studies to date have addressed the implications of the role of TLR-2 and TLR-4 in the skin of immunocompetent and immunosuppressed mice post *Acanthamoeba* spp. infection. Thus, to clarify the role of Toll-like receptors in the skin from *Acanthamoeba* infected host in relation to the immune status, the goal of this study was to determine (by two different methods, such as quantitative real time and immunohistochemistry) whether *Acanthamoeba* spp. may affect the expression and activity of Toll-like receptors in the skin.

## 2. Material and Methods

### 2.1. The Design of Experimental Model of Disseminate Acanthamoebiasis

The design of the experiment (mouse infection and animal immunosuppression) has been described in detail in our previous works [10,38]. Briefly, Balb/c mice (n = 96) were used in all experiments, and procedures were approved by Local Ethics Committee for Experiments on Animals in Szczecin (No. 29/2015, dated 22 June 2015) and Poznań (No. 64/2016, dated 9 September 2016). All efforts were made to minimize mice suffering.

The *Acanthamoeba* spp. trophozoites (AM 22 strain, genotype T16) used for the experimental infection were obtained from the bronchoaspirate of the patient [39]. Male mice were grouped into four groups as follows: A, amoeba-infected host with normal immunity (n = 30); AS, amoeba-infected host with reduced immunity induced by MPS, methylprednisolone sodium succinate (n = 30); C, control host with normal immunity (n = 18); CS, control host with reduced immunity induced by MPS (n = 18). For four days before amoeba infection, mice from AS and CS groups were intraperitoneally administered MPS (10 mg/kg in 0.1 mL of 0.9% NaCl) (Solu-Medrol, Pfizer, Puurs, Belgium, Europe), an anti-inflammatory glucocorticoid with immunosuppressive properties according to the methodology described by Markovitz et al. (1978) [40]. Animals from groups A and AS were intranasally given 3 μL of a suspension of *Acanthamoeba* trophozoites containing 10,000–20,000 of the amoebae counted in the Bürker chamber. Mice from the C and CS groups received 3 μL of 0.9% NaCl (dropped on the mouse’s nose). At different time points after infection (8, 16, 24 days post-infection, dpi), mice were euthanized with a peritoneal overdose of sodium pentobarbital (Euthasol vet, FATRO) (2 mL/kg body weight) to obtain tissues that were adequately secured.

The mice were subjected to daily observation. Twenty-four infected animals with dermatological changes, including alopecia, thinning and tousled fur and coat color changes were qualified for the study of immune mechanisms occurring in the skin; 12 mice from both A and AS groups and 5 mice from both C and CS groups (total n = 34) qualified. During the autopsy, skin sections (2 cm × 2 cm) were taken from the upper left part of the mouse’s back. The skin was shaved, divided into two sections, and then preserved for histological examination in 4% formalin and for biochemical-immunological studies at −80 °C, respectively [41].

The scheme of the experiment is presented in Figure 1.

### 2.2. Quantitative Real-Time PCR (qPCR)

TLR2 and TLR4 gene expression in the skin of mice in different groups was measured by qPCR (performed on a Light Cycler instrument from Roche Diagnostic GmbH, Mannheim, Germany) as described previously [11,12].

Briefly, SYBR Green I was used for detection dye, the housekeeping gene PBGD (porphobilinogen deaminase) as the reference gene for mRNA quantification and two tested TLR genes were examined and presented below as primers. The amounts of TLR2 and TLR4 transcripts in each sample were determined by the geometric means of the transcript level of the housekeeping gene, and finally, the relative level of expression of the two tested genes was calculated on the basis of the E-method formula (Roche Diagnostic).

In this study the following primers were used: 

***TLR2*** [13] 

5′-AAA GAT GTC GTT CAA GGA GG-3′ 

5′-ATT TGA CGC TTT GTC TGA GG-3′ 

***TLR4*** [13] 

5′-TTC TTC TCC TGC CTG ACA CC-3′ 

5′-CTT TGC TGA GTT TCT GAT CCAT-3′

***PBGD*** [13]

5′-TGG ACC TAG TGA GTG TGT TG-3′

5′-GGT ACA GTT GCC CAT CTT TC 3′

### 2.3. Immunohistochemistry

The dissected samples of skin (sample size about 1 cm^3^, localization: thigh skin, number n = 5 from each group: C, A, AS and CS) were fixed in 4% formalin (Chempur, Poland) for at least 24 h and then washed with absolute ethanol (Standlab, Poland; 3 times over 3 h), absolute ethanol with xylene (Supelco, Merck, Germany; 1:1; twice over 1 h) and xylene (3 times over 20 min). Then, after 3 h of saturation of the tissues in liquid paraffin, the samples were embedded in paraffin blocks. Using a microtome (Microm HM340E), 3–5 μm serial sections were taken and placed on polysine microscope slides (Thermo Scientific, UK). The sections of the skin were deparaffinized in xylene and rehydrated in decreasing concentrations of ethanol and microwaved in citrate buffer (pH 6.0) to induce epitope retrieval. After slow cooling to room temperature, slides were washed in PBS twice for 5 min and then incubated with primary antibodies overnight (4 °C). Immunohistochemistry was performed using specific primary rabbit polyclonal antibodies against TLR2 and TLR4 (Santa Cruz Biotechnology, Inc., Dallas, TX, USA; cat. no. sc-10739 and sc-30002, respectively) in a final 1:500 dilution, after checking and standardized other recommended by manufacturer concentrations from 1:50 to 1:500. Sections were stained with an avidin-biotin-peroxidase system with diaminobenzidine as the chromogen (Dako Envision +Dual Link System- HRP (DAB+) and performed according to staining procedure instructions included. Sections were washed in distilled H_2_O and counterstained with hematoxylin. The IHC reaction with 1:500 concentrated antibodies was performed 3 times. For negative control, specimens were processed in the absence of primary antibodies. For positive control, samples of spleen were immunostained as described above. Positive staining of the skin sample was defined by visual identification of brown pigmentation using a light microscope (Leica, DM5000B, Wetzlar, Germany). The samples were independently examined by two experienced histologists. The score of immunoexpression was conducted according to immunoreactivity based on visual examination of intensity and is as follows: negative (−), almost negative (+/−), weak positive (+), moderate positive (++) and strong positive (+++).

### 2.4. Statistical Analysis

Statistical analysis was carried out using StatSoft Statistica v.10.0. Initially, the Shapiro–Wilk test was performed which revealed whether the distribution of the data followed a normal distribution. Since the data did not follow a normal distribution then nonparametric tests were used for statistical analysis. The Mann–Whitney U test was used to compare two groups, and the Kruskal–Wallis H test was used to compare three groups. A statistical significance point of *p* < 0.05 was adopted. GraphPad 4.0 software was used to create graphs.

## 3. Results

### 3.1. TLR2 and TLR4 Genes Expression

Statistical analysis of TLR2 gene expression in different time points in A and AS groups were statistically insignificant (H = 4.26, *p* = 0.24 and H = 5.44, *p* = 0.14, respectively). TLR2 gene expression was at similar levels in the A group at all time points and the C group. In the amoeba-infected hosts with reduced immunity group (AS), we found upregulation of TLR2 gene expression at 8 dpi compared to the CS group. The difference turned out to be statistically insignificant, probably due to the large standard deviation in the AS 8 dpi group. In the AS 16 and 24 dpi groups, gene expression was at a level similar to CS group. It was also shown that immune status did not affect the TLR2 gene expression (A vs AS) (Figure 2).

The TLR4 gene expression at different time points in A and AS groups was statistically insignificant (H = 6.55, *p* = 0.09 and H = 0.08, *p* = 0.99, respectively). In the amoeba-infected host with normal immunity group (A), we found statistical upregulation of the TLR4 gene expression at 8 dpi compared to the C group (U = 1.00, *p* < 0.05). In AS 16 and 24 dpi groups gene expression was at a level similar to C group. In the amoeba-infected host with reduced immunity (AS), we noted that TLR4 gene expression was at a similar level in the AS group at all time points and CS group. Taking into account the host’s immune status, the TLR4 gene expression was statistically higher in the skin of amoeba-infected mice with normal immunity (A) than in amoeba-infected hosts with reduced immunity (AS) at the beginning of the infection (Figure 3).

### 3.2. Results of Immunohistochemistry

Immunoexpression of TLR2 in the skin of uninfected immunocompetent mice (Figure 4A) was visible in some areas of connective tissue of the dermis (red arrow) and in some keratinocytes of the epidermis (white arrow); this expression was at a moderate/low level. After eight days following the *Acanthamoeba* spp. Infection, the level of immunoreactivity for TLR2 became higher (Figure 4C, red arrows), and gradually on the following days post-*Acanthamoeba* spp. infection it became lower (Figure 4E,G) and was visible only in the deeper connective tissue of the dermis or near the epidermis (red arrows).

Immunoexpression of the TLR2 skin of uninfected immunosuppressed mice (Figure 4B, red arrows) was much lower than in the immunocompetent control (Figure 4A). After *Acanthamoeba* spp. infection of immunocompetent mice, the level of these receptors slightly increased in the first week after *Acanthamoeba* spp. infection (Figure 4D, red arrows) and became similar to the uninfected immunocompetent mice at 16 and 24 dpi (Figure 4F,H, red arrows).

In the skin of uninfected immunocompetent mice, the immunoexpression of TLR4 (Figure 4I) is very weak and mainly located just under the epidermis (red arrows). In uninfected immunosuppressed mice skin (Figure 4J), TLR4-expression was higher than in immunocompetent mice, and it was visible in the deeper layer of connective tissue of the dermis, just near some hair follicles (red arrows). Similarly, TLR4 expression increased at the beginning of *Acanthamoeba* spp. infection (Figure 4K, red arrows), and then gradually became much lower (Figure 4M,O, red arrows) and very similar to uninfected immunocompetent mice (Figure 4I). The level of TLR4 immunoexpression in the skin of *Acanthamoeba* spp. infected immunosuppressed mice regardless of the duration of the infection was similar (Figure 4L–P, red arrows) to the level of uninfected immunosuppressed animals (Figure 4J).

## 4. Discussion

This study aimed to investigate, to our knowledge for the first time, the expression of TLR2 and TLR4 in skin immunocompetent and immunosuppressed mice post-*Acanthamoeba* spp. infection. These studies showed change (but not statistically significant) in the expression of TLR2 in uninfected and *Acanthamoeba* spp. Infected mice according to immunological status. However, we observed that TLR4 expression was significantly upregulated in the skin at eight days post-*Acanthamoeba* spp. infection in an immunocompetent host. The important thing for our studies was the expression of TLR4 in the immunocompetent host (significantly higher at eight days post *Acanthamoeba* infection) and immunoexpression of TLR4 in the skin (the area of connective tissue of the dermis) were observed on the same day. Our results were confirmed by two different methods (qPCR and immunohistochemistry) that TLR4 is stimulated in response to *Acanthamoeba* spp. infection in the skin problems. In our previous study, Kot et al. in the kidneys noted a higher TLR2 expression in immunosuppressed mice at 24 day post *Acanthamoeba* spp. infection, similarly in the heart there was upregulation but in immunocompetent mice [10]. In the eye, increased expression of TLR2 and TLR4 in immunocompetent mice was observed at eight, sixteen and twenty-four days post-*Acanthamoeba* spp. infection, but immunosuppressed mice showed significant differences in the expression of TLR2 at 16 and 24 dpi [11]. In the lungs of immunocompetent *Acanthamoeba* spp.-infected mice, it was observed that TLR2 expression was higher than TLR4 expression, and the expression of TLR2 and TLR4 was increased from two to thirty days post-*Acanthamoeba* spp. infection [13]. In brain immunocompetent mice, *Acanthamoeba* spp. infection was observed as having increased expression of TLR2 and TLR4 at the beginning of infection [12].

The study of Ordeix et al. confirmed the expression of TLR2 receptors in large mononuclear cells in the skin of dogs infected with *Leishmania infantum* [42]. Similarly, studies by Pereira-Fonseca et al. confirmed the higher expression of TLR2 and TNF-α in dogs infected with *Leishmania* [43]. In a study, Polari et al. demonstrated higher expression of TLR2 and TLR4 in monocytes in human cutaneous leishmaniasis (*Leishmania braziliensis*) [44]. Furthermore, higher expression of TLR2 and TLR4 was confirmed in experiments by Campos et al. in cutaneous leishmaniasis (*Leishmania braziliensis* and *Leishmania amazonensis*) [45]. In this study, TLR 2 and TLR4 expression was visible in the deeper layer of connective tissue of the dermis mice *Acanthamoeba* spp. infected.

Cutaneous acanthamoebiasis is a disease that is difficult to diagnose. Moreover, there is no effective method of treatment of this pathogen, and despite the use of complex mixed long-term therapies, it may result in the patient’s death [46]. So far, it is not clear whether *Acanthamoeba* is the direct cause of the skin symptoms, or whether they are a consequence of the amoeba’s presence in other organs [47]. The results presented by Hernandez-Jasso et al. showed that the skin of *Acanthamoeba*-infected mice exposed to UV-B radiation is a source of infection from which trophozoites can travel by blood to other organs [4]. Other results indicated that patient treatment with immunosuppressants after kidney transplantation led to the development of painful nodules on the skin [6]. This case confirmed that cutaneous acanthamoebiasis is present frequently in patients treated with immunosuppressants.

The skin is the first barrier in the contact with pathogens and is an important element of the non-specific defense reaction; therefore, the presence of TLRs in the skin and their roles are very interesting subjects for scientific research [47].

TLRs have been implicated in the pathogenesis of skin diseases, such as atopic dermatitis and psoriasis. The molecules have been shown to be important in cutaneous host defense mechanisms against common bacterial, fungal and viral pathogens in the skin [48]. TLR2 mediates the innate immune response to bacterial pathogens and induces Th1 cytokine secretion. TLR2 was differentially expressed at several time points, and most highly downregulated in crusted scabies at the beginning of infection [49]. The relationship between TLRs and skin *Acanthamoeba* spp. infection remains relatively unknown.

Clinical observations confirm that primary cutaneous acanthamoebiasis has a different outcome and prognosis depending on whether there is CNS involvement. In patients without CNS involvement, the onset of amoebic skin changes is acute to subacute, with multiple lesions continually developing. Conversely, patients with brain *Acanthamoeba* spp. infection generally develop cutaneous lesions as a late manifestation of systemic disease which carries a very poor prognosis and is almost universally fatal [50]. In this study, we used a strain with neurophilic effects where numerous amoeba trophozoites were re-isolated from brain fragments from *Acanthamoeba* spp.-infected immunocompetent and immunosuppressed mice [38].

In summary, our study confirmed for the first time the change in expression of TLR4 in the skin of mice infected with *Acanthamoeba* in immunocompetent hosts. TLR4 is upregulated in the skin of immunocompetent mice in response to *Acanthamoeba* spp. at the beginning of the invasion. Further understanding of the mechanism of action of this receptor may lead to the discovery of a new method of prevention and effective treatment of acanthamoebiasis.

## Figures and Tables

**Figure 1 pathogens-12-00631-f001:**
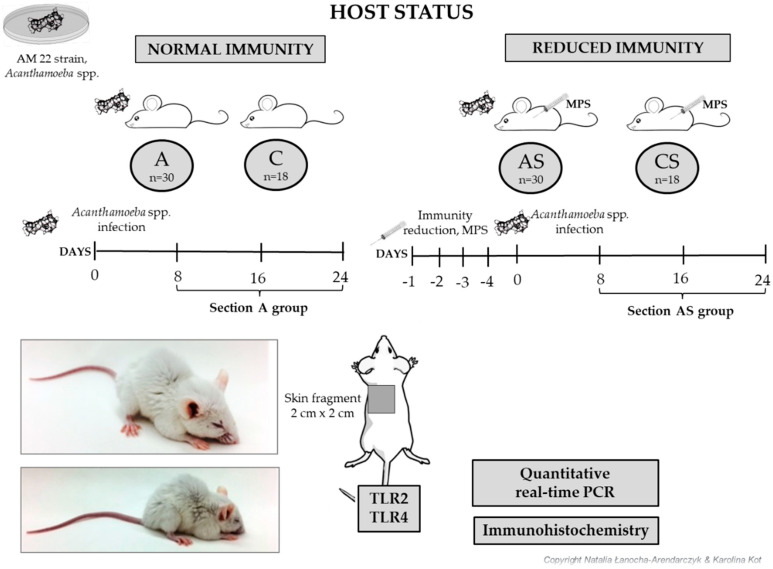
Schematic illustration of disseminated acanthamoebiasis experiment. Hosts were divided into two groups depending on their immunological status. Firstly, for four days before amoeba infection, mice assigned to AS and CS group were given methylprednisolone sodium succinate (MSP) for immunosuppression. Then, A and AS groups were inoculated the *Acanthamoeba* spp. trophozoites. After 8, 16 and 24 days post-infection (dpi), the mice were euthanized and the shaved skin samples were collected for analysis A, amoeba-infected host with normal immunity (n = 30); AS, amoeba-infected host with reduced immunity induced by MPS, methylprednisolone sodium succinate (n = 30); C, control host with normal immunity (n = 18); CS, control host with reduced immunity induced by MPS (n = 18). Original photos.

**Figure 2 pathogens-12-00631-f002:**
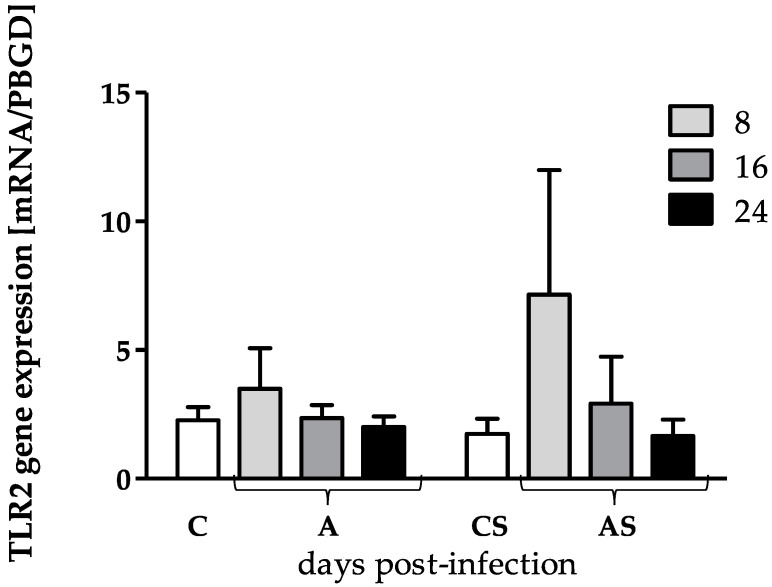
The TLR2 gene expression in the skin of control and amoeba-infected mice at 8, 16 and 24 dpi, (A, amoeba-infected host with normal immunity; AS, amoeba-infected host with reduced immunity induced by MPS, methylprednisolone sodium succinate; C, control host with normal immunity; CS, control host with reduced immunity induced by MPS). In the graph arithmetic mean ± standard deviation was shown; *p* < 0.05.

**Figure 3 pathogens-12-00631-f003:**
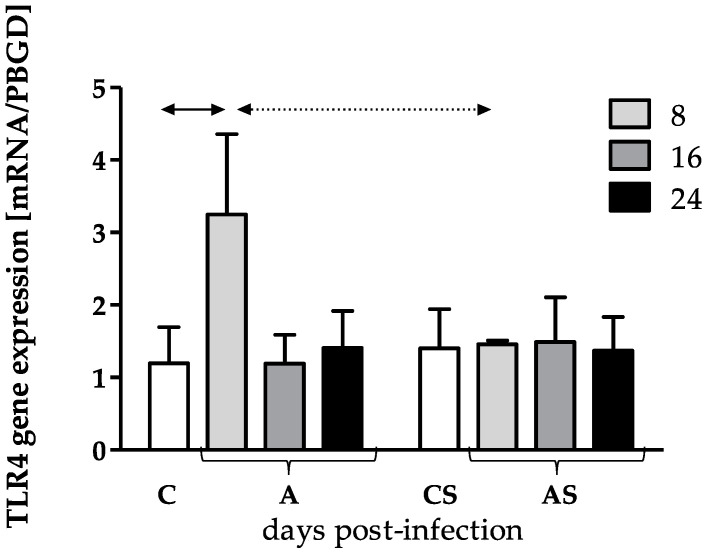
The TLR4 gene expression in the skin of control and amoeba-infected mice at 8, 16 and 24 dpi, (A, amoeba-infected hosts with normal immunity; AS, amoeba-infected hosts with reduced immunity induced by MPS, methylprednisolone sodium succinate; C, control hosts with normal immunity; CS, control hosts with reduced immunity induced by MPS). In the graph, arithmetic mean ± standard deviation was shown; *p* < 0.05; black arrow, difference between A vs. C; dotted arrow, difference between A vs. AS.

**Figure 4 pathogens-12-00631-f004:**
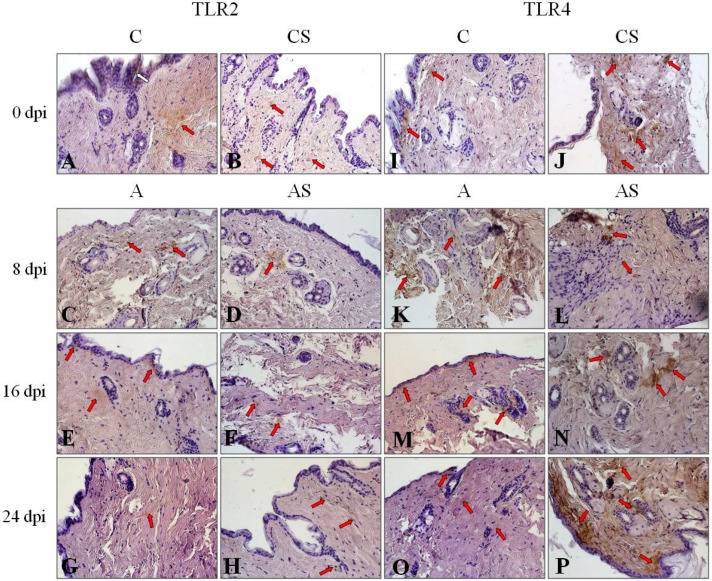
Representative microphotography that shows immunoexpression of TLR2 (**A**–**H**) and TLR4 (**I**–**P**) in skin of immunocompetent and immunosuppressed uninfected mice (0 dpi) and at 8, 16, and 24 days after *Acanthamoeba* spp. infection (dpi). Immunohistochemical reaction with diaminobenzidine (DAB) as a chromogen. Objective magnification 40×. Red arrows mark the area of connective tissue of dermis where immunoexpression of Toll-like receptors 2 (TLR2) and 4 (TLR4) was visible as brown pigmentation; white arrows mark the keratinocytes of epidermis where immunoexpression of TLR2 was visible as brown pigmentation. Abbreviation: C—uninfected immunocompetent mice (n = 5), CS—uninfected immunosuppressed mice (n = 5), A—immunocompetent *Acanthamoeba* spp. infected mice (n = 5), AS—immunosuppressed *Acanthamoeba* spp. infected mice (n = 5).

## Data Availability

Not applicable.

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
