# Peer review of "The Immunological Changes in the Skin of BALC/c Mice with Disseminated Acanthamoebiasis"

_pathogens, 2023, doi:10.3390/pathogens12050631_

Round 1

Reviewer 1 Report

The manuscript by  Wojtkowiak-Giera et al. entitled, “Cutaneous acanthamoebiasis –  the changes of TLR2 and TLR4 in the skin of Acanthamoeba spp. infected immunocompetent and immunosuppressed mice” is aimed to determine the mRNA expression of tlr2 and tlr4 receptors in the skin of BALC/c mice infected with Acanthamoeba with AM55 strain isolated from a patient with acute myeloid leukemia (AML) and atypical pneumonia.

Overall, the manuscript is well-written, well-structured, and highly novel. The general readers will be interested in the summaries and findings. Publication would benefit the scientific community since these findings have potential implications for the discovery of a new method of prevention and effective treatment of acanthamoebiasis.

Some minor issues to improve:

Abstract

Line 26-27: Rephrase "The expression of the Toll-like receptors tested didn’t change in the mice treated with the immunosuppressive drug."

Suggest adding the graphical abstract.

Keyword: Suggest including "Acanthamoeba spp.", "TLR-4", and discard "immunohistochemical staining".

Introduction

Line 50-51: Some authors observed that Acanthamoeba inducted skin tissue necrosis [6]. There is only one reference. Please include all the relevant authors.

Line 59-60:  Trypanosoma brucei and Leishmania major should be written as T. brucei and L. major, respectively.

Line 68-70: Leishmania major and Trypanosoma cruzi should be written as L. major and T. cruzi, respectively.

The author should justify the selection of TLR-2 and TLR-4 in this study.

Line 99-100: briefly add the method to determine the expression and activity of TLR-2 and TLR-4.

Materials and Methods

Line 103-107: Suggest combining these two paragraphs.

Line 110- please provide the number of BALB/c mice used in your study. 

Line 122: The author should mention detecting dye used in this study.

Line 129-137: Please provide the amplicon size of each primer pair.

Results

Line 158: Subheading, suggest changing to "Expression of TLR2 and TLR4 Genes by qPCR"

Figure 1: Missing TLR4 mRNA expression description in the figure legend. Please add.

Line 175: Suggest changing the subheading "Results of Immunohistochemical Staining" to "Immunohistochemistry".

Discussion

In the first paragraph, the author should provide the essential interpretation based on the key findings and include a main piece of supporting evidence.

In the following paragraph, the author can compare and contrast the finding to the previous studies. 

Author Response

Dear Reviewer

Thank you very much for all your comments regarding our manuscript. I kindly inform you that your comments have been taken into account by us.

Abstract

Line 26-27: Rephrase "The expression of the Toll-like receptors tested didn’t change in the mice treated with the immunosuppressive drug. - This rephrase was modified.

Keyword: Suggest including "Acanthamoeba spp.", "TLR-4", and discard "immunohistochemical staining". -  It was changed.

Introduction

Line 50-51: Some authors observed that Acanthamoeba inducted skin tissue necrosis [6]. There is only one reference. Please include all the relevant authors. - References was added.

Line 59-60:  Trypanosoma brucei and Leishmania major should be written as T. brucei and L. major, respectively. - It was changed.

Line 68-70: Leishmania major and Trypanosoma cruzi should be written as L. major and T. cruzi, respectively. - It was changed.

The author should justify the selection of TLR-2 and TLR-4 in this study.  It was changed.

Line 99-100: briefly add the method to determine the expression and activity of TLR-2 and TLR-4. - It was added

Materials and Methods

Line 103-107: Suggest combining these two paragraphs. - We changed whole paragraph.

Line 110- please provide the number of BALB/c mice used in your study. - It was provided.

Line 122: The author should mention detecting dye used in this study. – It was modified and detecting dye was added.

Line 129-137: Please provide the amplicon size of each primer pair. It was changed. 

Results

Line 158: Subheading, suggest changing to "Expression of TLR2 and TLR4 Genes by qPCR" - It was changed.

Figure 1: Missing TLR4 mRNA expression description in the figure legend. Please add. – It was added.

Line 175: Suggest changing the subheading "Results of Immunohistochemical Staining" to "Immunohistochemistry". - It was changed.

Discussion

In the first paragraph, the author should provide the essential interpretation based on the key findings and include a main piece of supporting evidence.

In the following paragraph, the author can compare and contrast the finding to the previous studies. 

  • Modified structured discussion agree with suggestion reviewer.

Best regards

Monika Derda

Reviewer 2 Report

The article reports the expression of toll-like receptors TLR2 and TLR4 in cutaneous acanthoamebiasis of experimentally infected mice.

Comments in detail can be found in the pdf file. Please read them carefully.

1.       In the introduction, more information is needed about the parasite, life cycle, clinic aspects of cutaneous acanthamoebiasis and the role of toll-like receptors as a key part of the innate response in this disease.

2.       A great deal of information is needed on materials and methods. Please read carefully in the comments throughout the pdf file. The authors should explain why they did not identify the specie of Acanthamoeba.

3.       In the results;

     Check the legend of the figures. There is missing information.

Even if the values are not statistically significant, reference should be made to them. In particular, in figure 1, at 8 dpi in the AS- TLR2 group, the standard deviation value was high and that particular value seems to be twice from the other samples. The authors should explain the use of Mann-Whitney U or use a similar one to test for such differences.

In figure 2, there is an increase in TLR2 in the non-immunosuppressed control group on day 0 but there are no qPCR results in the same group and no immunohistochemistry of the control groups after day 0. The authors should have done qPCR and immunohistochemistry in the control groups on the same days post-infection as the infected groups. They must therefore explain why they did not do so or, on the contrary, incorporate such data.

4.       Discussion. Please check carefully every comment in this section in the pdf file.

Some information on the ligands on the parasite would also be useful in the discussion.

I found no discussion on the comparison between qPCR and immunohistochemistry results, which is needed. The authors point out that there were no changes in the receptors in the immunosuppressed animal samples (which is true from day 0 to 24), but in figure 1B, at 16 dpi in the AS group, the values were higher even than the immunocompetent group at the same time. That needs to be explained. If there is no changes on TLRs in immunosuppressed mice, why there are no TLR2 brown spots on immunohistochemical figure in immunosuppressed control mice?

5.       References. Please reduce the number of non-relevant references and self-citations and add those related to the comments. There are also highlighted references in pdf file.

 There is a kind of problem with the size of some words in pdf file. Please check it.

As Acanthamoeba infections is of great interest, the authors should highlight the importance of studying toll-like receptors in the immune responses against the parasite in the skin. As far as I can see, in this work there is the problem with authors who divide the same work into several parts and, in order not to fall into a possible misconduct, they fail to incorporate information that is needed to understand the paper. As I was lacking relevant information, mainly on materials and methods, I looked for related works and from there my opinion. Some of the authors have an article published in Parasite & Vectors (Kot et al. Parasites Vectors (2020) 13:480 https://doi.org/10.1186/s13071-020-04351-4) on these receptors in kidney and heart and it is evident that it was done with the same animals. In fact, they stated that there are 96 animals in the trial but there were 34. 96 was the number of animals in the previously published paper.

Author Response

REVIEWER 2

Dear Reviewer

Thank you very much for all your comments regarding our manuscript. I kindly inform you that your comments have been taken into account by us.

Answers for your questions:

  1. The introduction was modified.
  2. We provided the missing information about the protocol of the experiment. We described the route of infection, mouse autopsy, the condition of the skin and how skin samples were collected. Only 24 infected mice had dermatological changes, that’s why only them were classified into the immunological analysis of the skin. That’s why there was a divergence in the number of animals in the previous version of the manuscript.

Natalia Łanocha-Arendarczyk from Pomeranian Medical University in Szczecin isolated the AM22 strain in 2007 from the patient during the diagnostic process.  In the same year genotyping of the AM22 strain was performed. In Poland, only genotypes are important in treatment of Acanthamoeba spp. infection that’s why the specie of Acanthamoeba was not identify. Thank you for the advice. We agree that it will be really interesting to know the specie that’s why it will be our future goal.

  1. In the results we provided the missing information.

We changed the results section, including the graphs. Initially, the Shapiro-Wilk test was performed, which revealed whether the distribution of the data followed a normal distribution. Since the data did not follow a normal distribution then nonparametric tests were used for statistical analysis. The Mann-Whitney U test was used to compare two groups (A vs C; AS vs. CS, and A vs. AS), and the Kruskall Wallis H test was used to compare three groups (8 dpi vs. 16 dpi vs. 24 dpi).

We agree with the Reviewer that data distribution give the very large dispersion and it’s a limitation of a study. The pathological process occurring within the host as a result of the infection by Acanthamoeba spp. is not fully understood. Many studies on acanthamoebiasis in experimental animals have reported individual differences in susceptibility to invasion, hence a possible source of data variability (Martinez and Kasprzak 1980). Even in experimental conditions, parasite models involve several often uncontrolled parameters, including the relationship between route of administration, timing and duration of immunosuppressive treatment, mouse lineage, host–parasite interactions, and type of immune response (Chatelain and Konar 2015).

In figure 2, there is an increase in TLR2 in the non-immunosuppressed control group on day 0 but there are no qPCR results in the same group and no immunohistochemistry of the control groups after day 0. The authors should have done qPCR and immunohistochemistry in the control groups on the same days post-infection as the infected groups. They must therefore explain why they did not do so or, on the contrary, incorporate such data.

Two methods we done control groups in the same days post -infection as the infected groups .
In all figure this group is improve and separate.

  1. Discussion was improved agree with suggesting reviewer

  1. We explained the diverge in number of mice in previous point and also we provided all information in the main text.

Reviewer 3 Report

In this study, the authors investigated the expression of TLR2 and TLR4 in the skin of Acanthamoeba spp. infected immunocompetent and immunosuppressed mice. Using qPCR and immunohistochemical staining, they provided evidence that the tlr4 receptor in the skin is involved in the host's immune defense triggered during Acanthamoeba infection. Although the findings are interesting, more in-depth analyses are needed. Main comments are as follows:

1.  It is not clear to me why the authors only tested the expression of TLR2 or TLR4. Does Acanthamoeba infection trigger TLR2 or TLR4 pathway activation? How about other TLRs?

2. In Fig.1B, the TLR4 mRNA expression in the skin of uninfected (0 dpi) looks similar in A and AS groups. However, TLR4 expression was higher in uninfected immunosuppressed mice skin (Fig. 2J) than in uninfected immunocompetent mice skin (Fig.2I). These data are inconsistent. Can the authors explain why?

Author Response

REVIEVER 3

Dear Reviewer

Thank you very much for all your comments regarding our manuscript. I kindly inform you that your comments have been taken into account by us.

Answers for your questions:

  1. It is not clear to me why the authors only tested the expression of TLR2 or TLR4. Does Acanthamoeba infection trigger TLR2 or TLR4 pathway activation? How about other TLRs?

In the first step we selected TLR2 and TLR4, because is little known about signaling pathway of acanthamoebiasis. In our earlier studies these  two receptors on others organs  and confirmed  changes in the expression of TLR2 and TLR4 therefore selected these receptors.
In the future, we will plan to examine other receptors.

  1. In Fig.1B, the TLR4 mRNA expression in the skin of uninfected (0 dpi) looks similar in A and AS groups. However, TLR4 expression was higher in uninfected immunosuppressed mice skin (Fig. 2J) than in uninfected immunocompetent mice skin (Fig.2I). These data are inconsistent. Can the authors explain why?

 These data may be different because  we check the expression TLR4  of two level  (mRNA and protein).  qPCR is method quantitative measurement of gene copy numbered Immunohistochemistry is method only check and localize specific antigens in sample of tissue.

Best regards

Monika Derda

Round 2

Reviewer 2 Report

Dear authors,

the manuscript has been substantially improved. I have no other remarks, but please check in the manuscript a few yellow highlightings about spacing and syntax errors that should be corrected. I also suggest removing in the title the word hosts as being too generic and add the type of host, i.e. Balb/c mice.

Adding figure 1 was a good decision!

 Best regards.

Author Response

Dear Reviewer
Thank you very much for taking the time to review our manuscript, your valuable comments and for your kindness.

I kindly inform you that some errors occur when converting a WORD file to a PDF. These errors are generated by the computer system, e.g. red fonts, unnecessary tabs, etc.

Best regards
Monika Derda

Reviewer 3 Report

The revised manuscript is substantially improved.

Author Response

Dear Reviewer
Thank you very much for taking the time to review our manuscript, your valuable comments and for your kindness.

Best regards
Monika Derda